# High-Cholesterol Diet Decreases the Level of Phosphatidylinositol 4,5-Bisphosphate by Enhancing the Expression of Phospholipase C (PLCβ1) in Rat Brain

**DOI:** 10.3390/ijms21031161

**Published:** 2020-02-10

**Authors:** Yoon Sun Chun, Sungkwon Chung

**Affiliations:** Department of Physiology, Sungkyunkwan University School of Medicine, Suwon, Gyeonggi-do 16419, Korea; ysun129@skku.edu

**Keywords:** phosphatidylinositol 4,5-bisphosphate, phospholipase C, cholesterol, high-cholesterol diet

## Abstract

Cholesterol is a critical component of eukaryotic membranes, where it contributes to regulating transmembrane signaling, cell–cell interaction, and ion transport. Dysregulation of cholesterol levels in the brain may induce neurodegenerative diseases, such as Alzheimer’s disease, Parkinson disease, and Huntington disease. We previously reported that augmenting membrane cholesterol level regulates ion channels by decreasing the level of phosphatidylinositol 4,5-bisphosphate (PIP_2_), which is closely related to β-amyloid (Aβ) production. In addition, cholesterol enrichment decreased PIP_2_ levels by increasing the expression of the β1 isoform of phospholipase C (PLC) in cultured cells. In this study, we examined the effect of a high-cholesterol diet on phospholipase C (PLCβ1) expression and PIP_2_ levels in rat brain. PIP_2_ levels were decreased in the cerebral cortex in rats on a high-cholesterol diet. Levels of PLCβ1 expression correlated with PIP_2_ levels. However, cholesterol and PIP_2_ levels were not correlated, suggesting that PIP_2_ level is regulated by cholesterol via PLCβ1 expression in the brain. Thus, there exists cross talk between cholesterol and PIP_2_ that could contribute to the pathogenesis of neurodegenerative diseases.

## 1. Introduction

Lipid metabolism including cholesterol, oxysterols, fatty acids, and phospholipids is involved in numerous neurodegenerative diseases including Alzheimer’s disease [1,2]. Cholesterol is a key component of plasma membrane bilayers, where it affects structural and physical functions including fluidity, curvature, and stiffness. It also regulates the functions of membrane proteins and is involved in transmembrane signaling processes, membrane trafficking, endocytosis, and ion transport [3,4]. Cholesterol also participates in the biosynthesis of bile acids and steroid hormones in the plasma membrane, which in turn play important functional and biological roles as signal transducers [5]. Cholesterol is particularly enriched in the brain, and the central nervous system contains 23% of all cholesterol in the whole human body. Cholesterol in neurons and astrocytes controls synaptic transmission [6]. Thus, changes in cholesterol levels and homeostasis have been associated with brain diseases and neurodegenerative disorders such as Alzheimer’s disease (AD), Parkinson’s disease (PD), and Huntington’s disease [7].

Phosphatidylinositol 4,5-bisphosphate (PIP_2_) plays a pivotal role in cell membranes, regulating biological functions including signal transduction, membrane trafficking, transporter functions, and ion channels [8,9]. Phospholipase C (PLC), which is grouped into four major families (β, γ, δ, and ε), hydrolyzes PIP_2_ to produce inositol 1,4,5-trisphosphate and diacylglycerol [10]. These products induce calcium release from intracellular stores and activate phosphorylation of cAMP response element-binding protein and neuronal gene expression in the brain [11,12]. Decrease of PIP_2_ in the cell membrane results in impaired long-term potentiation and reduced cognition [13]. Levels of PIP_2_ are closely related to the production of β-amyloid peptide (Aβ), a culprit in AD [14]. Many ion channels are regulated by PIP_2_, and the hydrolysis of PIP_2_ by phospholipase C has been shown to reduce their activities [8]. We demonstrated that augmenting membrane cholesterol levels regulated ion channels by decreasing PIP_2_ [15], suggesting that a close relationship between plasma membrane-enriched lipids, cholesterol, and PIP_2_. The human ether-a-go-go related gene (HERG) K^+^ channel, which is known to be modulated by PIP_2_, is inhibited by cholesterol enrichment via increased phospholipase C (PLCβ1) expression [16]. Consistent with this, we showed that increasing cholesterol levels in cultured cells increased PLCβ1 and PLCβ3 expression levels among PLC isoforms (β1, β2, β3, β4, γ1, and γ2) and increased PLCβ1 expression induced the decrease of PIP_2_ levels [17]. These results suggest that there may exist cross talk among two plasma membrane-enriched lipids, cholesterol and PIP_2_, via expression of PLCβ1. However, these results obtained from cultured cells are not confirmed in the brain yet.

In this study, we tested whether cholesterol enrichment affected PLCβ1 expression and PIP_2_ level in brain. We found that PIP_2_ levels decreased significantly in the cerebral cortices of rats on a high-cholesterol diet, repeating our previous in vitro result [15]. The high-cholesterol diet slightly increased expression of PLCβ1 but not that of PLCβ3. Levels of PLCβ1 expression correlated with those of PIP_2_, whereas levels of cholesterol and PIP_2_ did not. These results could suggest that PIP_2_ levels are regulated by cholesterol via PLCβ1 expression in the brain.

## 2. Results

### 2.1. PIP_2_ Levels Were Down-Regulated in Rats on a High-Cholesterol Diet

We previously found that enriching membrane cholesterol decreased PIP_2_ in cultured cells [14,16]. In order to test whether cholesterol augmentation in the brain affects levels of PIP_2_ in vivo, 13-week-old Sprague Dawley (SD) rats were fed with either normal or high-cholesterol diets for 6 weeks [18,19,20,21]. We observed that the body weight was not altered in high-cholesterol diet rats compared to normal diet rats (data not shown). First, we tested the levels of free cholesterol in the cerebral cortex. For this purpose, we obtained the membrane fractions from the cerebral cortices and measured the levels of free membrane cholesterol using assay kits. As shown in Figure 1A, the free cholesterol level was 390 ± 29 μg/mL (*n* = 10) in the rats on the normal diet, whereas it increased by 15% to 450 ± 44 μg/mL (*n* = 10) in the rats on the high-cholesterol diet. Cholesterol levels was increased by 15.4% by high-cholesterol diet. However, the difference between the two groups was not statistically significant (*p* = 0.24). This was likely owing to the varying cholesterol levels within each group as shown by scattered data in Figure 1A. Alternatively, the non-significant difference could have been because of the existence of different pools of cholesterol in the central nervous system. We next analyzed the PIP_2_ levels in the cerebral cortex membranes using assay kits. As shown in Figure 1B, PIP_2_ levels were 5.81 ± 0.34 pM (*n* = 10) in the rats on the normal diet, but they significantly decreased by 12.9% to 5.06 ± 0.26 pM (*n* = 10) in the rats on the high-cholesterol diet. These data indicate that PIP_2_ was down-regulated by cholesterol augmentation in vivo.

### 2.2. PLCβ1 Expression Increased in Rats on a High-Cholesterol Diet

The hydrolysis of PIP_2_ mediated by PLC is the major catabolic pathway for PIP_2_ [10]. Because we found that increased cholesterol levels led to increased PLCβ1 expression in the cultured cells [17], we examined whether high-cholesterol diet in rats affected expressions of PLCβ1 and PLCβ3. Levels of PLC expressions were measured from the cytosol and membrane fractions of the cerebral cortices using Western blot analysis. A typical Western blot for PLCβ1 and PLCβ3 is shown in Figure 2A. Full images of Western blot are shown in Appendix A. All Western blot bands for PLCβ1 in membrane and cytosol fractions are shown in Appendix A. We confirmed equal protein loading by β-tubulin levels, and the relative band densities of PLCβ1 and PLCβ3 compared with β-tubulin are shown in Figure 2B, C (*n* = 10). PLCβ1 expression slightly increased in membrane fractions from rats on the high-cholesterol diet although the effect was not statistically significant (Figure 2B; *p* = 0.512). The high-cholesterol diet did not affect the PLCβ1 expression in the cytosol fraction, and the PLCβ3 expression in both the cytosol and membrane fractions showed no differences between the two groups (Figure 2C).

### 2.3. PLCβ1 Expression Directly Correlated with PIP_2_ Levels

We compared the PLCβ1 and PLCβ3 expression along with PIP_2_ levels from individual rats. We hypothesized that if PLC expression directly affected PIP_2_, these two would correlate. As shown in Figure 3A, we compared PLCβ1 expression (shown as relative to tubulin expression) with PIP_2_ levels. There was a moderate correlation between them (RSQ, 0.297) from rat brains on the normal diet (open symbols, and a thin line). Interestingly, the PLCβ1 expression and PIP_2_ levels were more tightly correlated in the rats on the high-cholesterol diet (closed symbols and a thick line; RSQ, 0.345). These results suggested that even in the normal diet rats, PIP_2_ level in the brain correlated closely with PLCβ1 expression and that when cholesterol levels increased in high-cholesterol diet rats, the correlation and significance increased further. In the case of PLCβ3, however, there was no correlation between its expression levels and PIP_2_ levels in either the normal (open symbols; Figure 3B) or the high-cholesterol (closed symbols; Figure 3B) diet rats. Consistent with this, we previously showed that increasing cholesterol levels in cultured cells decreased PIP_2_ levels by increasing PLCβ1 expression, but not by increasing PLCβ3 expression [17].

We next compared PIP_2_ levels along with cholesterol levels. As shown in Figure 3C, there was no correlation between PIP_2_ and cholesterol levels in either the normal (open symbols) or the high-cholesterol (closed symbols) diet rats, indicating that cholesterol level may not directly affect PIP_2_. Together, these results may indicate that the increased cholesterol level in the brain slightly increased PLCβ1 expression, reducing levels of PIP_2_. The effect of the high-cholesterol diet on the expression levels was specific for PLCβ1 given that we found no correlation between levels of PLCβ3 and PIP_2_.

## 3. Discussion

In this study, we demonstrated that a high-cholesterol diet significantly decreased PIP_2_ levels and slightly increased PLCβ1 expression. The correlation between PIP_2_ level and PLCβ1 expression further increased in high-cholesterol diet rats. In the high-cholesterol diet rats, cholesterol increased by 15% compared with the rats on normal diet, although the difference was not statistically significant, likely because of the varying cholesterol levels in each group. These varying levels may also explain why there was no significant effect of the high-cholesterol diet on the PLCβ1 expression. Importantly, however, there existed a correlation between the expression and PIP_2_ levels even in the normal diet rats, and the correlation increased further in the high-cholesterol diet rats, indicating that the increased PLCβ1 expression by the high-cholesterol diet might have resulted in PIP_2_ hydrolysis. These results suggest that there exist relationships among cholesterol, PIP_2_, and PLCβ1 in the brain. We tested the effect of dietary cholesterol on levels of PIP_2_ and PLCβ1 in healthy young male rat. However, it was shown that different physiological conditions such as gender and age could affect a dysregulation in brain cholesterol metabolism [22,23]. In addition, cholesterol concentration in serum significantly differed in male and female rats fed the cholesterol-containing diet [24]. Thus, further experiments will be needed to compare the effect of a high-cholesterol diet on PLCβ1 expression and PIP_2_ levels by gender and age.

PIP_2_ is a minor lipid component in the plasma membrane, localized to the cytoplasmic leaflet of the phospholipid bilayer [25]. It plays important roles in the attachment of the cytoskeleton to the plasma membrane, endocytosis, membrane trafficking, and enzyme activation [9]. A previous report suggests that there exist different pools of PIP_2_ in the cell membrane to play its multiple roles [26]. PIP_2_ levels are determined through their production by phosphoinositide kinases and through their breakdown by phosphatases and phosphoinositide-specific PLC. Our results showed that PIP_2_ was closely related to PLCβ1 expression but there was no correlation with PLCβ3 expression, and these results are consistent with our previous report on cultured cells [17]. Taken together, we concluded that cholesterol enrichment specifically increased the expression of PLCβ1, resulting in the down-regulation of PIP_2_. Lipid raft may serve as a platform for the specific regulation of PLCβ1 expression by cholesterol enrichment, since it is known as a microdomain enriched with cholesterol [27,28]. PIP_2_ is shown to localize in lipid rafts [29,30,31]. Interestingly, PLCβ1 is also localized in lipid raft microdomains, when prepared from the synaptic plasma membrane fraction of rat brains [32]. However, further studies will be needed to clarify how cholesterol regulates PLCβ1PLCβ1 expression.

Cholesterol is essential for structural and physiological functions of neurons [4]. It has been suggested that changes in cholesterol homeostasis induce synaptic degeneration and the disruption of brain functions, contributing to neurodegenerative diseases [33]. Some studies showed that a high plasma cholesterol level increases the risk of developing PD [34], and the decrease of plasma cholesterol by statin might attenuate the deposition of α-synuclein in the brain [35]. The association of cholesterol with AD is one of the most studied topics [36]. Cholesterol levels are closely connected to the production of Aβ [37,38,39]. Also, cholesterol accumulates in senile plaques of AD patients and in transgenic APP(SW) mice [40]. Hypercholesterolemia has been shown to increase Aβ deposition and amyloid plaque formation in a transgenic mouse model [21]. Consistently, cholesterol synthesis inhibitors, statins, decrease Aβ production [41]. Since levels of PIP_2_ are closely related to the production of Aβ [14], PIP_2_ may serve as the important molecule that links cholesterol to the pathogenesis of neurodegenerative diseases such as AD.

## 4. Materials and Methods

### 4.1. High-Cholesterol Diet

Starting at 13 weeks of age, 10 male SD rats were placed on a high-cholesterol diet containing 5% cholesterol, 10% fat, and 2% sodium cholate for 6 weeks. A total of 10 animals were also placed on a normal diet containing 0.005% cholesterol and 10% fat for 6 weeks. The cerebral cortices were removed and immediately frozen and stored at −80 °C. All experiments on rats were carried out in accordance with the approved animal care and use guidelines of the Laboratory Animal Research Center in Sungkyunkwan University School of Medicine and all experimental protocols were approved by the Laboratory Animal Research Center in Sungkyunkwan University School of Medicine (#skkumed10-05, 10 January 2010).

### 4.2. Protein Extraction

The cerebral cortices were homogenized in Tris-buffered saline solution (20 mM Tris, 137 mM NaCl, pH 7.4) containing a protease inhibitor cocktail, and the extraction ratio (brain tissue: Tris-buffered saline) was 1:10 (*w*/*v*). Homogenate samples were sonicated for 1 min on ice and centrifuged at 1000× *g* for 10 min at 4 °C to remove nuclei and debris. Supernatants were separated into membranes (pellet) and cytosols (supernatant) by centrifugation at 100,000× *g* for 1 h at 4 °C. The pellet was lysed with Ripa buffer (25 mM Tris, 5 mM EDTA, 137 mM NaCl, 1% Triton X-100, 1% sodium deoxycholate, 0.1% SDS with a protease inhibitor cocktail, pH 7.4). The protein content was measured by Bradford assay (Bio-Rad Laboratories, Hercules, CA, USA).

### 4.3. Cholesterol Assay

We measured cholesterol using the Amplex Red cholesterol assay kit (Thermo Fisher Scientific, Halethorpe, CA, USA). Homogenate samples were centrifuged at 1000× *g* for 10 min to remove nuclei and cell debris. Membranes were pelleted from the supernatants by centrifugation for 1 h at 100,000× *g* at 4 °C and analyzed according to the supplier’s instructions.

### 4.4. Western Blot Analysis

We resolved protein from each sample on SDS-PAGE and transferred the resolved protein to nitrocellulose membrane. Membranes were blocked with 5% nonfat milk powder in Tris-buffered saline/Tween 20 (TBST) for 1 h at room temperature, followed by incubation with anti-PLCβ1 (Santa Cruz Biotechnologies, Dallas, TX, USA), anti-PLCβ3 (Santa Cruz Biotechnologies), and anti-β-tubulin (Sigma-Aldrich, St. Louis, MO, USA) antibodies for overnight at 4 °C; the dilutions were 1:500 for the PLC isozymes and 1:4000 for β-tubulin. After being washed with TBST, the membranes were incubated with horseradish peroxidase-conjugated goat anti-rabbit IgG (Invitrogen, Waltham, MA, USA) for 1 h at room temperature. We visualized the peroxidase activity with enhanced chemiluminescence and quantified the detected signals using the Fujifilm LAS-3000 system with Multi Gauge software (Tokyo, Japan).

### 4.5. PIP_2_ Assay

We measured the amount of PIP_2_ extracted from the membrane fractions of the homogenate using a PIP_2_ Mass ELISA kit (Echelon Biosciences Inc., Salt Lake City, CA, USA). We extracted the PIP_2_ from the normal and high-cholesterol diet groups according to the supplier’s instructions. We also estimated the cellular PIP_2_ quantities by comparing the values from the standard curve, which showed linear relationships at concentrations ranging from 0.5 to 1000 pM.

### 4.6. Statistical Analysis

Data were expressed as mean ± SEM. We performed statistical comparisons between controls and treated experimental groups using one-way ANOVA and considered *p* < 0.05 statistically significant. Box plot graphs was used to show the distribution of the observed data variation, which displayed as minimum to maximum value together with distribution around the median and 25th and 75th percentile as edges.

## Figures and Tables

**Figure 1 ijms-21-01161-f001:**
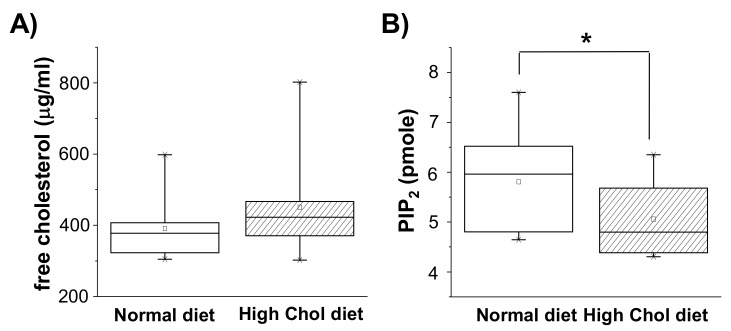
High-cholesterol diet decreased phosphatidylinositol 4,5-bisphosphate (PIP_2)_ levels in the rat cerebral cortex. Starting at 13 weeks of age, 10 male rats were placed on a high-cholesterol diet and 10 male rats were placed on a normal diet for 6 weeks as described in Materials and Methods. Cerebral cortex was removed and membrane and cytosol fractions were obtained. (**A**) The levels of free cholesterol in the membrane fractions were measured by cholesterol assay kit. Box plots show average cholesterol level in normal diet (open bar) or high-cholesterol diet (hatched bar) groups. The horizontal black lines represent the median of each distribution and the squares indicate means (*n* = 10). (**B**) PIP_2_ levels in the membrane fractions were measured by using PIP_2_ ELISA kit. The levels of PIP_2_ were decreased by a high-cholesterol diet (*n* = 10). * *p* < 0.05.

**Figure 2 ijms-21-01161-f002:**
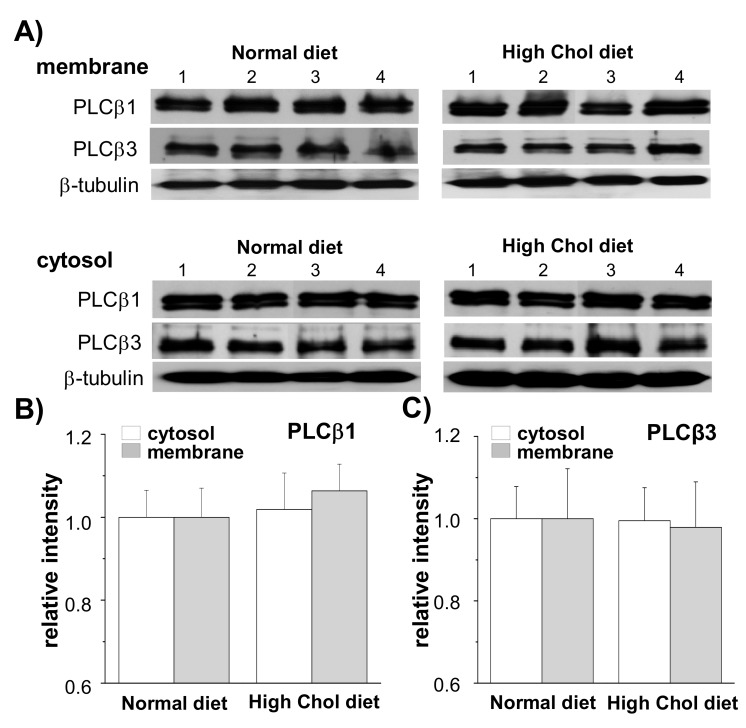
High-cholesterol diet slightly increased phospholipase C (PLCβ1) expression levels in the rat cerebral cortex. The expression levels of PLCβ1 and PLCβ3 were measured from the cytosol and membrane fractions of cerebral cortex using Western blot analysis. Representative Western blots are shown for PLCβ1 and PLCβ3. β-tubulin was used to confirm the amount of proteins loaded. (**A**) Expression levels of PLCβ1 and PLCβ3 were compared. (**B**,**C**) Bars indicate the levels of PLCβ1 (**B**) and PLCβ3 (**C**) obtained from densitometric analysis of Western bands in (**A**) (*n* = 10).

**Figure 3 ijms-21-01161-f003:**
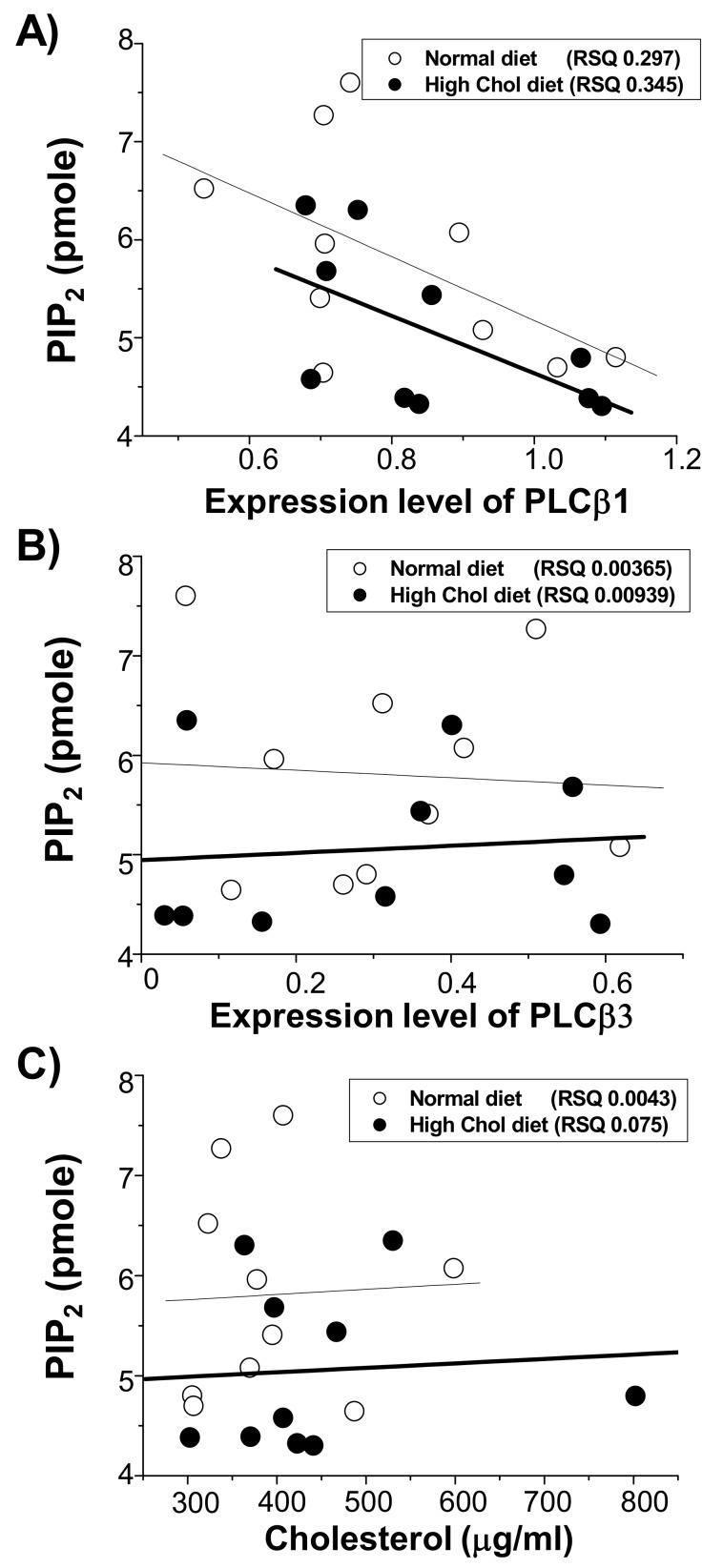
PIP_2_ levels correlated with the PLCβ1 expression levels in the rat cerebral cortex. PIP_2_ levels, PLCβ1 expression levels, PLCβ3 expression levels, and cholesterol levels from Figure 1 and Figure 2 were re-plotted. Linear regressions were performed to obtain RSQ values from rats on normal diet (open symbols and a thin line), and from rats on a high-cholesterol diet (closed symbols and thick line). (**A**) PIP_2_ levels correlated with the PLCβ1 expression levels. (**B**) There was no correlation between PIP_2_ levels and PLCβ3 expression levels. (**C**) There was no correlation between PIP_2_ levels and cholesterol levels.

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
