# Peer review of "High-Cholesterol Diet Decreases the Level of Phosphatidylinositol 4,5-Bisphosphate by Enhancing the Expression of Phospholipase C (PLCβ1) in Rat Brain"

_ijms, 2020, doi:10.3390/ijms21031161_

Round 1
Reviewer 1 Report
In this manuscript the authors intend to investigate whether high cholesterol diet in rats affects PIP2 and phospholipase C in brain. The experimental design is based on previous findings from the authors.
I have some concerns and comments that I would like to address to the authors:
1) How did the authors base their choice of 5 % of cholesterol in the diet composition? Previous findings have consistently shown that 0,15% to 0,5% is enough to establish hypercholesterolemia conditions in mice. The authors should justify their choice by referring to the literature.
2) It could be helpful to measure cholesterol levels in blood, if possible.
3) Levels of free cholesterol could be measure also in the cytosol, or in the whole homogenates from brain (levels of free cholesterol in the membrane are not significantly different in high cholesterol diet rats)
4) The authors state that PCLβ1 increased in rats on a high-cholesterol diet. Such statement is not supported by the results shown by the authors. PCLβ1 expression is not statistically increased in membrane fractions from rats on high chol diet! Moreover, the P value is not indicated in the text. The authors should measure mRNA levels of PCLβ1 and/or its activity.
Author Response
1) High-cholesterol diet can induce neuroinflammation and neuroapoptosis, resulting in neurodegeneration and cognitive deficits. As animal model for these high-cholesterol diet disorders, 5% cholesterol diet was used in previous reports. We added the following sentence and references in the results section (page 4, line 36).
“In order to test whether cholesterol augmentation in the brain affects levels of PIP2 in vivo, 13-week-old Sprague Dawley (SD) rats were fed with either normal or high-cholesterol diets for 6 weeks [18-21].”
2) Since dysregulation of cholesterol levels in the brain has shown to induce neurodegenerative diseases, we were interested in changes of PIP2 levels. Unfortunately, we did not take blood from rats.
3) Cholesterol is a key component of plasma membrane bilayers and 40-90% of total cellular cholesterol exists in the plasma membrane (Liscum and Munn, 1999). Thus, we anticipated large changes in free cholesterol in the plasma membranes by high-cholesterol diet. In addition, since PIP2 plays a pivotal role in cell membranes, regulating biological functions including signal transduction, membrane trafficking, transporter functions, and ion channels [6, 7], we focused on cross talk between two plasma membrane, cholesterol and PIP2, via expression of PLCβ1 in the membrane.
4) As Reviewer suggested, we changed description that the PCLβ1 expression is slightly increased in membrane fractions from rats on high cholesterol diet. Also, P value was indicated in the text.
We corrected the following sentence in the results section (page 5, line 34).
“PLCb1 expression slightly increased in membrane fractions from rats on the high-cholesterol diet although the effect was not statistically significant (Fig. 2B; p=0.512).”
Liscum, L.; Munn, N.J. Intracellular cholesterol transport. Biochimica et Biophysica Acta - Molecular and Cell Biology of Lipids 1999, 1438, 19-37.
We previously performed semi-quantitative RT-PCR for PLCβ1 in HERG-transfected HEK293 cells [14]. mRNA level of PLCβ1 was increased by cholesterol treatment for 0.5 and 1 h, indicating that the level of PLCβ1 was increased by the up-regulation of transcription. PLC hydrolyzes PIP2 to produce inositol 1,4,5-trisphosphate (IP3) and diacylglycerol. Thus, PLCβ enzymatic activity can be measured by using purified PLCβ on artificial phospholipid vesicles containing the substrate, PIP2, and quantities the amount of [3H]-IP3 released from the vesicle (Ghosh and Smrcka, 2004). However, we confirmed that high cholesterol only increased the levels of PLCβ1 and PLCβ3 among PLC isoforms and the effects of cholesterol enrichment on HERG channel and Aβ production were prevented by inhibiting PLCβ1 expression [14, 15]. Thus, we tested whether high-cholesterol diet induces the change of PLCβ1 and PLCβ3 expression levels in this study.
Ghosh, M.; Smrcka, A.V. Assay for G Protein-Dependent Activation of Phospholipase C β Using Purified Protein Components. Methods Mol Biol. 2004, 237, 67-75.
Reviewer 2 Report
This is a serious work. The data are clearly presented and the paper is well written. Some modifications are however required before publication.
The authors asked whether cholesterol enrichment affected PLCβ1 expression and PIP2 level in brain
A - Data obtained: It was found that PIP2 levels significantly decreased in the cerebral cortices of rats with a high-cholesterol diet. The high-cholesterol diet increased PLCβ1 expression. The levels of PLCβ1 expression correlated with those of PIP2, whereas levels of cholesterol and PIP2 did not. These results could suggest that PIP2 levels are regulated by cholesterol via PLCβ1 expression in the brain.
Remarks at the end of the abstract I think it must be written: “These results could suggest that PIP2 levels are regulated by cholesterol via PLCβ1 expression in the brain” instead of “These results suggest that PIP2 levels are regulated by cholesterol via PLCβ1 expression in the brain”
Point 1: It is clear that PIP2 is slightly but significantly decrease
Point 2: It is also clear that PLCbeta1 (whereas not significant) is slightly increased in the cerebral cortices of rat with a high cholesterol diet for 6 weeks
Point 3: The most important is the excellent correlation of PLCbeta1 with PIP2 levels
I think that these three points must be underlined at the beginning of the discussion. The data could be briefly summarized before to discuss.
B- Introduction: Overall, the introduction is good. I however think that more information should be provided on the different isoforms of PLC in the brain. The paper focuses on PLC beta1, what about the other isoforms? This must be briefly introduced. In addition, it must be underlined that lipids, including phospholipids, but also others lipids (fatty acids, oxysterols…) play important roles in Alzheimer’s disease (AD) and that the study of lipid metabolism is crucial to better understand the physiopathology of AD. To underline this point, you could cite the following paper:
Zarrouk A, Debbabi M, Bezine M, Karym EM, Badreddine A, Rouaud O, Moreau T,
Cherkaoui-Malki M, El Ayeb M, Nasser B, Hammami M, Lizard G. Lipid Biomarkers in Alzheimer's Disease. Curr Alzheimer Res. 2018 Feb 22;15(4):303-312.
C-Results: The data are clearly presented. The results are honestly presented and seem sound. Data shown are supported by the supplemetary data.
D- Major criticisms: the following points must be discussed.
The authors used young rats 13 weeks old (at the beginning of the study) and 19 weeks old at the end (fat diet for 6 weeks), and only male. The data could be (may be) more significant with old rats. In addition, they use male. An experiment with old female rats could be also of interest since in human, AD is more frequent in female. I think it is important to discuss the model used.
What about the diet? Is it appropriated? With a diet with higher cholesterol content and for a longer period of time could we expect more pronounced differences between the groups (high fat diet (elevated cholesterol content) versus control (normal diet)?
The title must be modified:
High-cholesterol diet decreases the level of phosphatidylinositol 4,5-bisphosphate by enhancing the expression of phospholipase C (PLCβ1) in rat brain
Author Response
As Reviewer suggested, we corrected (page 2, line 24).
As Reviewer suggested, we added following sentences at the beginning of discussion section (page 6, line 42).
“In this study, we demonstrated that a high-cholesterol diet significantly decreased PIP2 levels and slightly increased PLCb1 expression. The correlation between PIP2 level and PLCb1 expression further increased in high-cholesterol diet rats.”
As Reviewer suggested, we added following sentences in the introduction section (page 3, line 56).
“Consistent with this, we showed that increasing cholesterol levels in cultured cells increased PLCβ1 and PLCβ3 expression levels among PLC isoforms (β1, β2, β3, β4, γ1, and γ2) and increased PLCβ1 expression induced the decrease of PIP2 levels [17].”
As Reviewer suggested, we added following sentences in the introduction section (page 3, line 4).
“Lipid metabolism including cholesterol, oxysterols, fatty acids, and phospholipids is involved numerous neurodegenerative diseases including Alzheimer’s disease [1, 2].”
To test whether dietary cholesterol affects levels of PIP2 in healthy young rat, 13-week-old SD rats were fed with high-cholesterol diets for 6 weeks. However, it was shown that different physiological conditions such as gender and age could affect to a dysregulation in cholesterol metabolism [22, 23]. Also, there are several reports about gender-based differences following high-cholesterol diet in rats. It was shown that cholesterol concentration in serum significantly differed in male and female rats fed the cholesterol-containing diet (2.48 μmol/mL vs 2.92 μmol/mL) [24]. High cholesterol diet-induced renal injury in female was higher than in male animals (Al-Rejaie et al., 2012). Thus, further experiments will be needed to compare the effect of a high-cholesterol diet on PLCb1 expression and PIP2 levels by age and gender.
As Reviewer suggested, we added following sentences in the discussion section (page 7, line 8).
“We tested the effect of dietary cholesterol on levels of PIP2 and PLCb1 in healthy young male rat. However, it was shown that different physiological conditions such as gender and age could affect a dysregulation in brain cholesterol metabolism [22, 23]. In addition, cholesterol concentration in serum significantly differed in male and female rats fed the cholesterol-containing diet [24]. Thus, further experiments will be needed to compare the effect of a high-cholesterol diet on PLCb1 expression and PIP2 levels by gender and age.”
Al-Rejaie, S.S.; Abuohashish, M.M.; Alkhamees, O.A.; Aleisa, A.M.; Alroujayee, A.S. Gender difference following high cholesterol diet induced renal injury and the protective role of rutin and ascorbic acid combination in Wistar albino rats. Lipids Health Dis. 2012, 11, 41.
We refer to several references to determine the cholesterol content and period of time. To induce hypercholesterolemia in a AD transgenic mouse model, animals were placed on a high-cholesterol diet containing 5% cholesterol [21]. Also, transgenic mice Tg2575 were fed with a 5% cholesterol diet for 6 weeks [20]. Thus, we decided high-cholesterol diet with 5% cholesterol for 6 weeks in rat.
As Reviewer suggested, we added the following references in the results section (page 4, line 36).
“In order to test whether cholesterol augmentation in the brain affects levels of PIP2 in vivo, 13-week-old Sprague Dawley (SD) rats were fed with either normal or high-cholesterol diets for 6 weeks [18-21].”
We changed the title as suggested.
Round 2
Reviewer 1 Report
Thank you for your answers and work.